# Description of *Candida auris* Occurrence in a Tertiary Health Institution in Riyadh, Saudi Arabia

**DOI:** 10.3390/healthcare11243150

**Published:** 2023-12-12

**Authors:** Fatimah S. Alshahrani, Abba Amsami Elgujja, Sara Alsubaie, Salah Ahmed Ezreqat, Ahmed M. Albarraq, Mazin Barry, Khalifa Binkhamis, Lulwa Alabdan

**Affiliations:** 1College of Medicine, King Saud University, Riyadh 11362, Saudi Arabia; falshahrani1@ksu.edu.sa (F.S.A.); mbarry@ksu.edu.sa (M.B.);; 2Division of Infectious Diseases, Department of Internal Medicine, King Saud University Medical City, King Saud University, Riyadh 11451, Saudi Arabia; 3IPAC Department, King Saud University Medical City, Riyadh 11362, Saudi Arabia; sezreqat@ksu.edu.sa; 4Pediatric Infectious Diseases Fellowship Program, College of Medicine, Internal Medicine (Pediatric Infectious Diseases) King Saud University Medical City, King Saud University and Consultant, Riyadh 11461, Saudi Arabia; salsubaie@ksu.edu.sa; 5Department of Pathology, College of Medicine, King Saud University, Riyadh 11362, Saudi Arabia; aalbarrag@ksu.edu.sa (A.M.A.); kbinkhamis@ksu.edu.sa (K.B.); 6Division of Infectious Diseases, Faculty of Medicine, University of Ottawa, Ottawa, ON K1H 8M5, Canada

**Keywords:** *Candida auris*, *C. auris*, candida, candidemia, multidrug-resistant organism, MDRO, emerging pathogens, resistant pathogens

## Abstract

Background: *Candida auris* is an emerging multidrug-resistant fungal pathogen that represents a current serious threat to healthcare settings. Objective: The objective was to determine the prevalence of *C. auris* in a Riyadh hospital since its initial detection in late 2019. Methods: Using an adapted risk assessment tool, we reviewed the charts and medical files of all suspected and confirmed cases of *C. auris* infections reported at King Khalid University Hospital, Riyadh, between November 2019 and December 2022. Anonymized data were retrieved in a pre-established datasheet and analyzed to determine the epidemiological characteristics of *C. auris* infections in our facility. We analyzed prevalence by age, gender, risk factors, and according to sampling source. Results: Of the 53 confirmed *C. auris*-positive cases during the study period, 33 (62%) were males. Their ages ranged between 15 and 98, with most positive cases occurring in those aged 50 and above. Only one of the confirmed cases was hospital-acquired. All patients had at least one risk factor, and urine samples yielded the greatest number of positive cases, while admission to healthcare facilities constituted the highest risk in our study. Conclusion: Establishing a local prevalence pattern could serve as a baseline/benchmark to compare with regional and international benchmarks.

## 1. Introduction

This study investigated the incidence of *Candida auris* in a tertiary-care teaching hospital located in Riyadh, Saudi Arabia. *C. auris* first emerged in 2009 [1] and has continued to cause hospital-acquired infections in individuals with compromised immune systems and has also been associated with persistent candidemia and high mortality rates globally.

*Candida auris* was first identified when it was found in the outer ear of a patient in Japan [2]. Since then, clinical cases of *C. auris* infections have been reported in other countries including South Korea [3], India [4], Pakistan [5], South Africa [6,7], and Canada [8], with a remarkable surge between 2018 and 2021 in the United States [9]. The first Saudi case was reported in 2018 [10] and was later followed by several reports from other parts of the country [11,12]. 

This fungal species is rapidly spreading worldwide, with several outbreaks [13] reported from five continents in recent years [4]. Between the years 2019 and 2020, *C. auris* has been reported in over 40 countries around the world. The Asian continent has the highest number (14) of countries in which *C. auris* has been reported, followed by Europe (13), and then trailed far behind by South America (4), Africa (4), North America (3), and Oceania (1) [14]. 

The surge in the incidence rates of candidemia has created an immense burden on public health, particularly among patients in intensive care units [5,7]. The mortality rate is higher among infected patients compared with those who were colonized [15]. *C. auris* is an opportunistic pathogen that can cause candidemia [16]; the people at risk of infection include immunocompromised patients; patients who have received organ transplants; diabetic patients; or patients with recent antifungal use, catheter use, and prolonged ICU stays [9]. Other risk factors include chronic kidney diseases, recent vascular surgery, or surgery within the previous three months [17,18]. 

*C. auris* started attracting considerable global attention due to its growing reports, transmission through health professionals, high rate of treatment failure, and multidrug resistance [19]. *C. auris* is increasingly becoming a threat to human health because of its intrinsic resistance to one or more classes of antifungal drugs [14,20]. However, other studies have noted that *C. auris* is less resistant to 5-fluorocytosine and caspofungin [21].

Whole-genome sequencing has identified geographically distinct *C. auris* genotypes, suggesting region-specific resistance and transmission patterns. [22] Therefore, the multidrug resistance of *C. auris* can be geographically categorized as Clade 1: South Asian; Clade II: East Asian; Clade III: African; Clade IV: South American; and recently, Clade V: Iranian [23]. Although the clades are attributed to specific geographical locations, mixed isolates may be found in a single location [22,24]. Accordingly, a study has documented the trans-border importation of *C. auris* by patients with recent exposure to healthcare services in another country where *C. auris* has been reported [8]. 

It is difficult to identify *C. auris* using the traditional fungal identification methods, [25] and they can lead to incorrect identifications [9,25]. Owing to their close genetic relatedness, *C. auris* is often reported as *C. haemulonii* [14,26] while using conventional identification systems like APIC20C, Vitek2YST, and BD Phoenix [27], and as *C. parapsilosis* using RapiID [28], thus necessitating additional testing methods with higher specificity to elicit species identification [27,29]. The misidentification of *C.* auris can potentially result in incorrect treatment or the delay of proper treatment with the increasing chance of fatality [30]. 

Similarly, there are currently no established *C. auris*-specific susceptibility breakpoints, but clinicians often rely on their expert opinion and previously established breakpoints for other related *Candida* species. Currently, there is no evidence of a relationship between microbiologic breakpoints and clinical outcomes [31].

Furthermore, some of the epidemiological distinctiveness of *C. auris* includes its swift transmission [25] and its resistance to conventional disinfectants [32,33]. Unfortunately, *Candida auris* is considered one of the most virulent environmental pathogens that are associated with hospital transmission [13]. It can survive on surfaces for a prolonged period [1]. Moreover, the recent isolation of *C. auris* from a natural aquatic habitat in India indicated that this fungus may also exist without a human host [34].

As *C. auris* is spread rapidly within healthcare settings, it has become imperative to monitor its virulence and devise appropriate treatment approaches [9] in view of the several hospital-associated transmissions reported globally [35,36,37]. Therefore, the early detection and implementation of infection control practices can potentially reduce the risks [18,32]. Consequently, understanding its epidemiology can significantly help in planning specific infection control measures for healthcare settings [38,39]. Accordingly, this study aims to provide a descriptive overview of the occurrence of *C. auris* in a Riyadh hospital since its initial detection in late 2019. 

## 2. Methodology

This was a retrospective study of patients with a *C. auris* infection at a tertiary healthcare institution during the period from November 2020 to the end of 2022. It was during this period that the hospital started recording additional *C. auris* cases (after our first case in November 2019), and specific infection prevention and control measures were adopted and implemented to minimize hospital-associated transmissions within the hospital.

### 2.1. Data Collection

Although no patient identifiable data were used for the reporting, Institutional Review Board Approval No. 22/0701/IRB was obtained before proceeding with the research. The ethical approval required us to abide by the rules and regulations of the Kingdom of Saudi Arabia and the research policies and procedures of the KSU IRB related to data privacy. 

Accordingly, all data were mined from the hospital’s electronic health information system (*eSIHI*) after properly anonymizing, i.e., after removing the patient’s medical record number, national identification number, nationality, names, and any other patient identifiers. The data on patients’ demographic information, baseline features, comorbidities, laboratory results, and clinical outcomes were then compiled in an Excel v. 2310 worksheet and analyzed after reviewing the electronic patient records.

### 2.2. Specimen Sampling

All *C. auris* strains identified in the lab throughout the research period from both clinical and surveillance screening samples were included. For the purpose of inclusion, all active surveillance, contact tracing samples, and clinical samples were considered. Surveillance samples include those taken when it was determined that the risk factors for colonization or infection existed, or if an inpatient was included as a contact of a positive case (as part of contact tracing). 

As can be seen in Table 1, the list of risk factors was selected from the findings of several previous studies. Each risk factor was given a score of 1, except for previous admission to another hospital that would have subjected the patient to a combination of numerous hospital-acquired risk factors, including MDROs, critical comorbidities, and hospital-acquired and device-related infections. A minimum score of 3 from the selected risk factors or a previous admission to other hospital(s) necessitated surveillance screening. Active surveillance samples included nasal, axilla, groin, wounds, in-dwelling device sites, etc. Clinical samples included those taken from patients whose clinical condition required investigations for *C. auris*. Such patients may have included those being investigated for septicemia or septic shock. Additionally, based on clinical assessment, other sites were potentially included like the anus, chronic wounds, blood, urine, wounds, tissues, drains, etc. For the purpose of this study, only the first positive isolate per patient was included.

### 2.3. Testing and Identification

Surveillance swabs were cultured on Sabouraud dextrose agar with chloramphenicol and incubated at 37 °C for 48 h. Any growth of yeast from surveillance samples underwent identification. Additionally, significant growth of yeast from clinical samples (e.g., blood, urine, wounds, etc.) was identified. Yeast identification from surveillance and clinical samples was performed using matrix-assisted laser desorption ionization time-of-flight mass spectrometry (biomerieux, Marcy-l’Étoile, France).

### 2.4. Determining Hospital Associated Transmission

An infection may be designated as hospital-acquired or community-acquired (present on admission) depending on when, during the hospitalization, it manifests itself. In addition, for the purpose of determining if a *C. auris* isolate is classified as hospital- or community-acquired, the study adopted the epidemiological definition of hospital-acquired infections coined by the National Health Safety Network of the USA [40]. The Network considers hospital-acquired infections to be any infection that is determined not to be “present on admission (POA)”. 

An infection is considered as POA if an element of the specific infection criteria manifests during the POA window period, i.e., 2 days prior to an inpatient admission, on the admission date, or the calendar day after admission. In other words, if an element (sign or symptom) occurs after the second calendar date, it is considered a hospital-acquired infection.

Therefore, if a sample is taken after the second calendar date of admission and the patient did not exhibit any symptoms before the third calendar date, the isolate was considered to be hospital-acquired.

### 2.5. Statistical Analysis 

Statistical analysis was performed using SPSS version 28 (IBM Co., Ltd., Armonk, NY, USA). The inclusion criteria for the multivariable analysis was based on epidemiological criteria derived from the selected risk factors, the patient’s demographic information, and the outcome of the disease. The categorical data are expressed as frequencies and percentages (%), with a two-tailed *p* value < 0.05 considered statistically significant. 

We calculated the CI of rates using statistical methods appropriate for count data, such as the Poisson distribution for rare events and the binomial distribution for binary outcomes. The specific formula for calculating the CI of a rate depends on the distribution and the nature of the data.

We determined the total number of male and female patients admitted to the hospital during the specified time period and calculated the total number of infections in each gender group (male and female). The infection rate was calculated by dividing the number of infections in each group by the total number of patients in that group. The infection rates are expressed as percentages by multiplying the rates by 100.

## 3. Results

A total of 53 patients (33 males and 20 females) with *C. auris* infections were included in this study. Urine specimens were the most frequently obtained sample in 30.2% of patients followed by axilla samples (11.3%) and then thigh and anal specimens (9.4% each). The samples labeled as buttocks and hip were combined with those of the thigh; thus, the proportion of thigh samples would be 18.8% (n = 10), which is the highest after the urine samples.

### 3.1. Occurrence by Age

The median age of the screened patients was 64 years (inter-quartile range (IQR) 15–98). A further analysis of *C*. *auris* incidence by age (Table 2) showed a drastic increase in the incidence of *C. auris* among patients aged 51 years and above, with a staircase-like increase with every additional 10 years above 51.

### 3.2. Occurrence by Patient Characteristics

Additionally, most patients (83%) had comorbidities, while half (50.9%) of them had been previously admitted to other hospitals. Other risk factors included admission to high-risk units (35.8%), wounds (34%), and use of in-dwelling (32.1%), as summarized in Table 3.

### 3.3. Occurrence as Hospital- versus Community-Acquired Infection

Out of the 53 isolates identified in this study, only 1 met the epidemiological definition of hospital-acquired *C. auris* infection/colonization. In other words, as per the NHSN surveillance definition of hospital-acquired infections alluded to in the methodology section, all but one was identified as being present on admission (POA) during active surveillance screening [40]. One case, considered hospital-acquired, was on admission for longer than one week when the patient’s condition worsened and a clinical sample tested positive. 

### 3.4. Occurrence as Infection versus Colonization

Out of the 53 isolates included in this study, 7 were isolated from clinical specimens taken when the patient’s condition worsened and required ICU admission. Four out of the seven that required ICU admissions developed *C. auris* candidemia. Therefore, out of the 53 cases, 4 were considered *C. auris* infections and the remaining 49 were considered colonization. Colonization is defined as the presence of pathogens without manifestations of an infection (signs and symptoms).

### 3.5. Occurrence as Clinical versus Surveillance Samples

Our sample included 7 clinical samples and 46 active surveillance samples. This highlights the vigor with which we endeavored to identify all possible patients that were at risk of *C. auris* infection/colonization for the purpose of proactively identifying and isolating all positive cases. The proportions of clinical versus surveillance samples could identify already sick (potentially infected) patients who were not necessarily sick but may have been colonized and were picked during the surveillance sampling. This could be useful in distinguishing our prevalence from other comparable hospitals. For instance, in a recent Saudi Arabian study [11] where 27 patients with invasive candidemia were studied for their risk factors and mortality, their prevalence was very different from ours as we had a greater number of colonized patients and only one infected patient.

## 4. Discussion

Our study’s strength comes from the fact that it is the first in Saudi Arabia to give verifiable proof of the prevalence of this infamous yeast in hospital settings. It seems evident our findings echo the conclusions from several other studies that *C. auris* is already prevalent across the globe, although at different rates. For instance, a systematic review [41] showed that, from January 2019 to January 2021, several countries, including those in the Middle East, have reported a significant number of *C. auris* cases. Using data collected from nine studies, they reported the number of cases from several countries, with 71 in Kuwait [42], 29 in Oman (from two separate studies) [43,44], 35 in Saudi Arabia, 47 in Spain [45], 12 in Mexico [46] and Kenya [47], and 47 in the USA (from two studies) [48,49]. These totals included both *C. auris* candidemia and colonization. 

Most of the patients (clinical and active surveillance) that tested positive for *C. auris* were males (33; 62%) (Table 2). This is consistent with the findings of a retrospective analysis of the clinical characteristics of *C. auris* infection worldwide from 2009 to 2020 [39] and another study where 62% of the patients were male [50].

The first three cases of *C. auris* were reported in Saudi Arabia in the year 2018 [10], and more cases were subsequently reported in other parts of the Kingdom [51,52,53]. The United Arab Emirates also reported its first case of *C. auris* candidemia in the same year as the Kingdom of Saudi Arabia [54]. Six other Middle Eastern countries also followed suit. All these patients, all adults, were also initially misdiagnosed as having *C. haemulonii* infections [27]. 

It is noteworthy that the first case of *C. auris* was reported in our health facility in late November of 2019, and our prevention and control measures were overshadowed by the declaration of COVID-19 as a global pandemic in March 2020 [55]. A number of factors have complicated the treatment of COVID-19 co-infections with *C. auris* during the COVID-19 pandemic. Such factors include the multidrug-resistant nature of *C. auris* and their shared risk factors including co-morbidities, immunosuppressive states, and mechanical ventilator-dependent states [56]. Therefore, a superimposed *C. auris* infection in a COVID-19 patient could exacerbate the severity of secondary comorbidities, including severe lung injury and acute respiratory distress syndrome (ARDS), and heighten mortality rates among critically ill patients [57,58]. Another similarity between the two pathogens is that they are both found on patient care environmental surfaces, e.g., floors and air ducts, making transmission among ventilated patients easy [59].

During the first 6 months of the study period (November 2019 to April 2021), four (active infections and/or active surveillance) cases were reported. However, through the remaining three-quarters of the year 2021, there were between one and three cases reported monthly. This could be associated with the optimization of the *C. auris* identification and control measures during this period. March 2022 marked the peak of the graph, with nine cases being reported. This period marked the massive active surveillance screening of many patients that had unprotected contact with a *C. auris*-positive patient. There was then a steady staircase-like increase in cases from June to September 2022, until it finally declined to zero in December 2022 (see Figure 1).

As shown in Table 2, the incidence of *C. auris* according to age showed a sharp rise in the incidence at the age of 51 and older. There was also a staircase-like increase every 10 years after the golden jubilee. This is consistent with the findings in other reports where nearly half of the cases were in patients around the age of 70 years [60,61]. A similar Omani study has shown that, out of 108 patients, 40 (37%) were >65 years of age [62]. However, a South African study that compared, among other aspects, the age distribution of patients with candidemia caused by *C. auris* with other candida species, showed that the incidence was highest among neonates, followed by those in the 40–70 year age bracket, but lowest in the 10–20 years age range [63]. However, in contrast to our findings, there was a steady decline in the rates every 10 years of age after 50 years. The bottom line would seem to be that extremes of age are a significant risk factor for *C. auris*.

Among the *C. auris*-positive samples, our study showed that urine samples yielded the highest number of cases (30.2%), followed by samples taken from the thigh, buttock, and hip (18.8%) and axilla (11.3%). There was only one case of candidemia reported. Most samples were taken during enhanced active surveillance rather than as diagnostic samples in suspected clinical infections (Table 2).

Although the first case of *C. auris* was found in Japan in the ear (hence the name *auris* [2]), it has since been found in different parts of the body. In a study of 108 clinical samples with *C. auris* isolates, the most common samples were blood (38.9%), urine (36.3%), respiratory (8%), central line tip (8%), wound (6.2%), and other samples (2.6%) [62]. It has also been found in bronchoalveolar lavages [64], diabetic foot tissue cultures, etc. [65], which is consistent with the findings in several other studies of clinical samples. In [66], it is noteworthy that this study involved mostly clinical samples taken from sick patients, while our study included mostly surveillance samples with quite a few clinical samples. Although they are distinctively different classes of microbes, *C. auris* cases are similar to MDROs in their risk factors, environmental source, and mode of transmission; in addition, patients infected with *C. auris* are often co-infected with other MDROs [8,67].

As shown in Table 3, a review of the patient characteristics revealed that all of the reported cases (100%) had at least one risk factor, similar to the findings in other studies where 98.1% had at least one risk factor [62]. In addition, 83% of our patients had co-morbidities that included chronic kidney diseases, septicemia, diabetes mellitus, or chronic lung disease. As reported in the results, 50.9% had been admitted to other hospitals, while 35.8% had been admitted to a high-risk unit. Approximately 34% of the patients had wounds, and 32.1% had devices including urinary catheters, central venous catheters, and mechanical ventilators. Our findings seem to reiterate the findings of other studies. In an Omani study, 68.5% had comorbidities [62], while a New York study showed that extensive healthcare exposure and underlying comorbidities constituted significant risk factors in the reported cases [61].

It is noteworthy that an alternative multiparametric approach may have been successfully used in some research, e.g., to identify high-risk prostate cancer (with a Gleason score of at least 7) with better sensitivity and specificity than that provided by PSA screening alone [68]. However, our approach is to identify and isolate potentially infected or colonized patients based on epidemiological criteria for the purpose of early recognition and isolation, and not on clinical criteria for treatment purposes.

However, this study is not without limitations. Being the first study of its type within the Kingdom, and with the lack of a regional benchmark to compare our rates with, it is difficult to present our prevalence rates in relative terms. Also, the use of risk factors for active surveillance screening might not be as specific as, for example, a multiparametric approach. We may have overlooked a few situations. 

The inherent risk of misdiagnosis or incorrect identification could result in missing some potential cases of *C. auris* colonization/infection. Although *C. auris* was properly identified, our hospital laboratory relied on the susceptibility of other candida species or on expert opinion in deciding on the appropriate antifungal treatment. This explains why we did not include the susceptibility results in our study.

## 5. Conclusions

*C. auris* is a public threat that is globally endemic and continues to spread within healthcare settings. *Candida auris* has become a major concern in healthcare due to the increasing number of immunocompromised patients. It is inherently resistant to disinfectants and antifungal therapies. 

It possesses the ability to survive in healthcare environments, rapidly colonize the skin of patients at risk, be easily spread within the healthcare setting, and hence potentially result in devastating and protracted outbreaks. Enhanced active surveillance for *C. auris* and infection control measures could avert such potential nosocomial outbreaks.

Following our index case in November 2019, our enhanced active surveillance suggests that C. *auris* may be under-reported and its endemicity in the Kingdom could be higher than expected. Our study has, further, shown that *C. auris* is more prevalent among immunocompromised patients, the elderly, patients with previous hospitalization, those admitted in critical or intensive care units, or those with chronic comorbidities.

However, several questions are still begging for answers regarding the epidemiology of *C. auris.* For instance, what makes it inherently resistant to multi antifungals? What enables *C. auris* to persist in clinical settings for long periods of time? Other questions include why it is seemingly more prevailed among male patients than female? Consequently, *C.* auris will continue to attract further research to help us better understand its resistance and pathogenic mechanism with a view to developing more specific therapeutic, preventive, and control measures.

## Figures and Tables

**Figure 1 healthcare-11-03150-f001:**
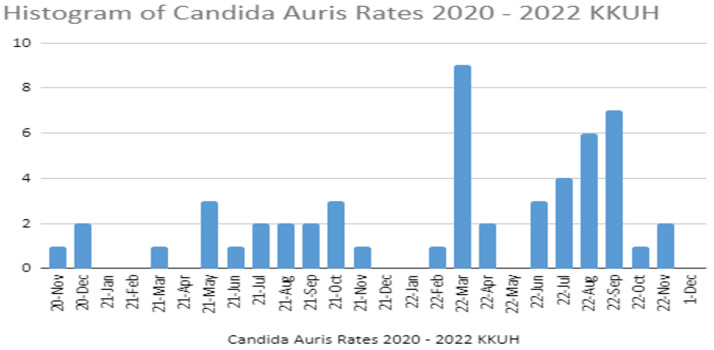
Histogram of *C. auris* rates in 2020–2022.

**Table 1 healthcare-11-03150-t001:** Risk factors for *C. auris* infection.

Bundle Element/Risk Factor	Score
1. History of admission from other hospital	3
2. Has any of these: septicemia + CKD, DM, or chronic lung disease	1
3. Previous history of MDRO infection or colonization	1
4. History of admission in hospital outside the KSA	1
5. Presence of wounds or indwelling devices	1
6. Admission to high-risk units (ICU, HDU, Oncology etc.)	1
7. Contact of MDRO/ASC	1
8. Previous surgery <3 months	1

HDU: high-dependency unit; ICU: intensive care unit; ASC: active surveillance culture; KSA: Kingdom of Saudi Arabia.

**Table 2 healthcare-11-03150-t002:** Patients’ characteristics (n = 53).

	N	%
Age (years)		
≤20	2	3.8
21–30	4	7.5
31–40	3	5.7
41–50	2	3.8
51–60	13	24.5
61–70	14	26.4
≥71	15	28.3
Gender		
Male	33	62.3
Female	20	37.7
Specimen		
Urine	16	30.2
Axilla	6	11.3
Thigh	5	9.4
Anus	5	9.4
Arm	4	7.5
Swab	3	5.7
Penis	3	5.7
Hip	3	5.7
Nose	2	3.8
Buttock	2	3.8
Leg	2	3.8
Neck	2	3.8
Nasal	2	3.8
Tissue	1	1.9
Wound	1	1.9
Nail	1	1.9
Rectal	1	1.9
Blood	1	1.9
Foot	1	1.9

**Table 3 healthcare-11-03150-t003:** Patients’ risk factors.

	N	%	95% CI of Rate
Comorbidities	44	83	60.3 to 111.5
Admission to other hospital	27	50.9	33.6 to 74.1
Admission to high-risk unit	19	35.8	21.6 to 56
Wounds	18	34.0	20.1 to 53.7
Devices	17	32.1	18.7 to 51.4
Antimicrobials	12	22.6	11.7 to 39.6
ASC	11	20.8	10.4 to 37.1
Surgeries	7	13.2	5.3 to 27.2
MDRO	1	1.9	0.0 to 10.5
Outside KSA	0	0.0	---
Contact of MDRO	0	0.0	---

CI: confidence interval.

## Data Availability

Raw study data are not readily available online but can be made available on request following local regulations and policies.

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
