# Peer review of "Description of Candida auris Occurrence in a Tertiary Health Institution in Riyadh, Saudi Arabia"

_healthcare, 2023, doi:10.3390/healthcare11243150_

Round 1

Reviewer 1 Report (New Reviewer)

Comments and Suggestions for Authors

The authors present descriptive data regarding the occurrence of Candida auris within a healthcare institution in Riyadh, as part of a retrospective study involving 53 patients. Candida auris infections have witnessed a significant global increase and pose a substantial health threat, particularly to susceptible patient populations. Hence, the importance of this subject necessitates an encouragement for publication in this field. 

Key issues identified include: 

  • The manuscript suffers from suboptimal English language usage, hindering comprehension. The abstract, in particular, requires greater clarity. For instance, the statement in line 29, "Only one of the confirmed cases was hospital-acquired," contradicts the claim in line 31 that "admission to healthcare facilities constituted the highest risk." It is essential to clarify that the authors are referring to healthcare facilities other than the one under study. 

  • The manuscript lacks meticulous proofreading, as evident in numerous typos, extraneous characters, and grammatical errors. The language quality should undergo thorough revision. 

  • In the results section (line 165), the authors amalgamate buttock, hip, and thigh specimens and provide the percentage for this combined group in relation to the total patients. The rationale behind this grouping and the additional insights gained from this approach remain unexplained. 

  • There is a lack of explanation regarding the scoring system employed for assessing risk factors in Table 1. 

Hence, it is advisable to conduct a comprehensive revision of the manuscript to address these concerns.

Comments on the Quality of English Language

The authors present descriptive data regarding the occurrence of Candida auris within a healthcare institution in Riyadh, as part of a retrospective study involving 53 patients. Candida auris infections have witnessed a significant global increase and pose a substantial health threat, particularly to susceptible patient populations. Hence, the importance of this subject necessitates an encouragement for publication in this field. 

Key issues identified include: 

  • The manuscript suffers from suboptimal English language usage, hindering comprehension. The abstract, in particular, requires greater clarity. For instance, the statement in line 29, "Only one of the confirmed cases was hospital-acquired," contradicts the claim in line 31 that "admission to healthcare facilities constituted the highest risk." It is essential to clarify that the authors are referring to healthcare facilities other than the one under study. 

  • The manuscript lacks meticulous proofreading, as evident in numerous typos, extraneous characters, and grammatical errors. The language quality should undergo thorough revision.

  • In the results section (line 165), the authors amalgamate buttock, hip, and thigh specimens and provide the percentage for this combined group in relation to the total patients. The rationale behind this grouping and the additional insights gained from this approach remain unexplained. 

  • There is a lack of explanation regarding the scoring system employed for assessing risk factors in Table 1. 

Hence, it is advisable to conduct a comprehensive revision of the manuscript to address these concerns.

Author Response

English editing is already done.

See attached responses

Reviewer 2 Report (New Reviewer)

Comments and Suggestions for Authors

Tables 2 & 3 have the same name, authors should put a title describing the included data

Characterization of the type of infection Hospital versus Community-Acquired C. auris or Infection versus Colonization, Clinical versus Surveillance samples methods should be clarified in the manuscript in detail. Also, results should be clarified in details

Comments on the Quality of English Language

can be modified

Author Response

All concerns have been addressed.

Please see the attached file for the comments

Reviewer 3 Report (New Reviewer)

Comments and Suggestions for Authors

Alshahrani and co-authors studied the incidence of Candida auris at their health institution. I agree with the authors that these studies are extremely important for the medical community, especially in the times of another on going pandemic, and appreciate their efforts in the preparation of first report from Saudi Arabia. I recommend publication of this report after the following major and minor corrections to the manuscript.

Major corrections:

·         Table 1: Please explain why only one of the risk factor is given a score of 3, while the rest are given only 1.

·         L160: On what data were the statistical analyses performed as none of this data was presented? Why did the authors choose two-tailed t-test? How does the data look with a one-tailed t-test, with the same alpha?

·         Table 3: What is the significance of CI of rate? It will be good to define ‘rate’ and how the CI was calculated.

·         L216-L219: The absolute number of patients and the percentages only provide partial information. It will be useful to normalize this data with the total number of male or female patients admitted to the hospital. The percentage of infections may be higher in men if more men were visiting the hospital (and vice versa for women).

·         Figure 1: The plot can be repasted properly as an image, instead of pasting the screenshot from Excel directly.

Minor corrections (grammatical/format etc):

·         L2: Candida auris (small a for auris, and the name in italics)

·         L39: At the beginning of the sentence, ‘the’ may be removed as C. auris is a noun.

·         L44: Repeated word may be deleted (in in).

·         L49, L78: The singular of species is species, and not ‘specie’. These sentences may be edited accordingly.

·         Tables will be much easier to read if the text is left aligned, and not center aligned.

·         L224: C. haemulonii (capital C)

·         L23, L244, L308: Question mark? should be removed.

·         L264: the sentence is incomplete as the n value (in the brackets) was not reported.

Comments on the Quality of English Language

Many sentences in the text are grammatically incorrect. It will be useful to get the manuscript revised to correct them and reduce complexity in their structure.

Author Response

All issues raised have been addressed.

Please see the attached file for details.

This manuscript is a resubmission of an earlier submission. The following is a list of the peer review reports and author responses from that submission.

Round 1

Reviewer 1 Report

Comments and Suggestions for Authors

1.       The introduction should start by providing a clear and concise statement of the research objective or main question that the article aims to address regarding Candida auris.

 2.       The authors should provide a brief overview of the global prevalence and distribution of Candida auris outbreaks, highlighting the countries affected and the magnitude of the problem.

 3.       It is important to clarify the significance of the emergence of Candida auris as a healthcare-associated infection and the specific challenges it poses in terms of infection control practices.

 4.       The article should provide a clear definition of multidrug resistance in the context of Candida auris and explain which antimicrobial agents are typically affected.

5.       When mentioning the difficulties in diagnosis and the potential for mismanagement of treatment, the authors should provide more information on the specific challenges and limitations of conventional laboratory methods in accurately detecting C. auris.

6.       The report of C. auris misdiagnosis as C. haemulonii in Middle Eastern countries should be supported by the relevant study or publication and provide details on the implications of such misdiagnosis.

 7.       The claim that C. auris presents distinctive epidemiological characteristics, including swift transmission, screening and detection, and susceptibility to conventional disinfectants, should be supported by specific examples or studies.

 8.       The authors should provide a rationale for the specific time period chosen for the study (from November 2020 to the end of 2022) and explain why this duration is appropriate for capturing relevant data on C. auris cases.

 9.       While patient identifiable data was not used for reporting, the authors should provide more details about the data anonymization and protection procedures implemented to ensure compliance with ethical guidelines and patient privacy.

10.   The inclusion criteria for C. auris strains identified in the lab should be clearly defined, including the sources of the clinical and surveillance screening samples, and any specific laboratory methods or criteria used for identifying C. auris.

 11.   The authors should provide more information on how patient demographic information, baseline features, comorbidity, laboratory results, and clinical outcomes were collected and recorded. The data collection process and any tools or forms used should be described in detail.

 12.   When using the methicillin-resistant Staphylococcus aureus (MRSA) risk assessment tool as a yardstick for identifying potential C. auris cases, the authors should provide a clear rationale for this choice and explain the similarities between C. auris and bacterial multi-drug resistant organisms (MDROs).

 13.   Authors should assess whether the assumptions for performing a Kaplan Meier analysis are met.

 14.   Several figures presented as a pie chart could be omitted and presented descriptively in the text.

 15.   Some figures from the survival analysis can be joined together and presented as a panel graph.

 16.   Explain what was the criteria for inclusion of variables in the Multivariable analysis. Was it a statistical or epidemiological criterion?

 17.   It is necessary to improve the discussion trying to contrast the results with those of other studies.

 18.   Authors are required to review and state some additional limitations to their study before conclusions.

Comments on the Quality of English Language

None.

Author Response

1.        

The introduction should start by providing a clear and concise statement of the research objective or main question that the article aims to address regarding Candida auris.

Addressed in the Introduction.

See page 2; Line 41-45

2.        

The authors should provide a brief overview of the global prevalence and distribution of Candida auris outbreaks, highlighting the countries affected and the magnitude of the problem.

See pp 2-3; Lines: 56 - 73

3.        

It is important to clarify the significance of the emergence of Candida auris as a healthcare-associated infection and the specific challenges it poses in terms of infection control practices.

P 6; Lines 149 - 162

4.        

  The article should provide a clear definition of multidrug resistance in the context of Candida auris and explain which antimicrobial agents are typically affected.

Pp 3-4; Lines 87 - 96

5.        

When mentioning the difficulties in diagnosis and the potential for mismanagement of treatment, the authors should provide more information on the specific challenges and limitations of conventional laboratory methods in accurately detecting C. auris.

Pp 5-6; Lines: 125 - 148

6.        

The report of C. auris misdiagnosis as C. haemulonii in Middle Eastern countries should be supported by the relevant study or publication and provide details on the implications of such misdiagnosis.

Pp 5-6; Lines: 125 - 148

7.        

The claim that C. auris presents distinctive epidemiological characteristics, including swift transmission, screening and detection, and susceptibility to conventional disinfectants, should be supported by specific examples or studies.

P 6; Lines 149 - 163

8.        

The authors should provide a rationale for the specific time period chosen for the study (from November 2020 to the end of 2022) and explain why this duration is appropriate for capturing relevant data on C. auris cases

P7; lines 192 - 196

9.        

While patient identifiable data was not used for reporting, the authors should provide more details about the data anonymization and protection procedures implemented to ensure compliance with ethical guidelines and patient privacy.

Pp 7-8; lines 197 - 211

10.     

The inclusion criteria for C. auris strains identified in the lab should be clearly defined, including the sources of the clinical and surveillance screening samples, and any specific laboratory methods or criteria used for identifying C. auris.

P 8; lines 212 - 224

11.     

The authors should provide more information on how patient demographic information, baseline features, comorbidity, laboratory results, and clinical outcomes were collected and recorded. The data collection process and any tools or forms used should be described in detail.

Pp 7-8; lines 197 - 211

12.     

When using the methicillin-resistant Staphylococcus aureus (MRSA) risk assessment tool as a yardstick for identifying potential C. auris cases, the authors should provide a clear rationale for this choice and explain the similarities between C. auris and bacterial multi-drug resistant organisms (MDROs).

Expunged, to be used in another study.

13.     

Authors should assess whether the assumptions for performing a Kaplan Meier analysis are met.

Expunged, to be used in another study.

14.     

Several figures presented as a pie chart could be omitted and presented descriptively in the text.

All figures have been described and cross-referenced.

All irrelevant figures have been removed.

15.     

Some figures from the survival analysis can be joined together and presented as a panel graph.

The survival aspect is removed, and kept for another relevant study.

16.     

Explain what was the criteria for inclusion of variables in the Multivariable analysis. Was it a statistical or epidemiological criterion?

The survival aspect is removed, and kept for another relevant study.

17.     

It is necessary to improve the discussion trying to contrast the results with those of other studies.

Addressed in pp14-17

18.     

Authors are required to review and state some additional limitations to their study before conclusions.

Reviewed and expanded. See p 17

Reviewer 2 Report

Comments and Suggestions for Authors

Title: The Prevalence of Candida Auris in a Tertiary Health Institution in Riyadh, Saudi Arabia

Respectfully and after a judicious exercise, my conclusion is that despite the fact that it is an interesting topic, it is poorly focused and analyzed. There are several inaccuracies in the concepts and the statistical analyzes shown are very basic and descriptive that do not provide new knowledge and ignore the complexity of the scenarios where mycosis is relevant.

Some of the impressions that I consider blur the paper are:

1.            C. auris is an opportunistic pathogen, the associated comorbidities are not explained in depth and I do not understand how the impact of the COVID-19 pandemic is unknown at the time of the study.

2.            The constant comparison with MDRO bacteria, and even more so with Staphylococcus aureus, puzzles me, it could be interesting, well documented and explaining the adjacent biological differences that categorically differentiate the two scenarios, just to mention that fungi, being eukaryotic cells, do not they transmit genetic material horizontally. Among other differences.

3.            During the introduction nothing is mentioned about the circulating clades of this yeast, nor about the epidemiology inherent to these clades, within which is the description of the associations with antifungal resistance.

4.            In the statistical analyzes, a multiparametric approach should be used, with individualized results, it is not known, for example, which individual had cancer, for example, was admitted to the ICU and suffered a candidaemia, everything is analyzed separately and it is impossible to understand the context.

5.            Therefore, I consider that the objective of the paper to describe the prevalence could be under or overestimated, and that the information that exists in the networks, and the actuality of this pathogen are not worked with the detail that it should have.

6.            In the lines (124-126), say: Susceptibility testing  was performed on clinical isolates using the Sensititre Yeast one Y010 AST Plate (Thermon  fisher, Massachusetts, United States). However, the results do not appear, and then there is a sentence that I did not have enough money. I don't quite understand what happened, but it is essential information to know the resistance profile of the isolates in this study.

 Essentially for these reasons, I consider that the paper cannot be published

Author Response

1.        

C. auris is an opportunistic pathogen, the associated comorbidities are not explained in depth and I do not understand how the impact of the COVID-19 pandemic is unknown at the time of the study

Addressed in pp 6-7; lines 163 - 177

2.        

The constant comparison with MDRO bacteria, and even more so with Staphylococcus aureus, puzzles me, it could be interesting, well documented and explaining the adjacent biological differences that categorically differentiate the two scenarios, just to mention that fungi, being eukaryotic cells, do not they transmit genetic material horizontally. Among other differences.

P 16; lines 395 - 398.

3.        

During the introduction nothing is mentioned about the circulating clades of this yeast, nor about the epidemiology inherent to these clades, within which is the description of the associations with antifungal resistance.

P4; lines 97 - 108

4.        

In the statistical analyzes, a multiparametric approach should be used, with individualized results, it is not known, for example, which individual had cancer, for example, was admitted to the ICU and suffered a candidaemia, everything is analyzed separately and it is impossible to understand the context.

P 10; lines 266 – 272

Multiparametric approach may have been successfully used in some researches, e.g., to identify high-risk prostate cancer, however, our approach is to identify and isolate potentially infected or colonized patients based on epidemiological criterion for the purpose of early recognition and isolation, and not on clinical criteria for treatment purposes.

5.        

In the lines (124-126), say: Susceptibility testing was performed on clinical isolates using the Sensititre Yeast one Y010 AST Plate (Thermon fisher, Massachusetts, United States). However, the results do not appear, and then there is a sentence that I did not have enough money. I don't quite understand what happened, but it is essential information to know the resistance profile of the isolates in this study.

The susceptibility aspect is removed

This phrase “I did not have enough money” could not be located in the scripts as it were.

Reviewer 3 Report

Comments and Suggestions for Authors

Authors describe a retrospective study of patients that were diagnosed with C. auris infection from tertiary healthcare institution. Objective of the study was to identify prevalence in Saudia Arabia with the intent to evaluate local precautionary measures and mitigate the spread of infection. Novelty seems to be in regional findings and implications more globally. 

Introduction: As currently written, it remains unclear how prevalence and hospital protocol (i.e., precautionary measures, spread of infection etc.) are linked. 

Consider expanding the introduction to describe the current clinical strategy for identifying C. auris and subsequent protocols for mitigating the spread of infection. What was/is the strategy for c. haemulonii , which according to the authors, is the organism misdiagnosed when evaluating for C. auris. How do these clinical strategies differ between different Arab nations (i..e, Saudia Arabia and The United Arab Emirates). 

Methods: clearly describe what software and/or protocols were used to retrospectively analyze the data. As currently written, it remains unclear if all the samples originated from Saudia Arabia or elsewhere. Remains unclear why only the first isolate per patient was included. Also, why were there yeast cultures performed; is the susceptibility data included? It is difficult to identify the corresponding data to each assay; indicating the figure or table after each method or result would be helpful. 

Results. consider including a more thorough description for each table and figure. As currently written, it is unclear why clinical versus surveillance samples is relevant to the original question and intent of the study. Need to describe all of the data in each figure. Seems authors selected only a subset of reported data to describe at length in the results section. 

Discussion: Perhaps the weakest portion of the manuscript. Several of the figures are not mentioned at all in the discussion and not referenced in text. Either expand the discussion to include all of the figures/data or remove several figures that are not mentioned or discussed. In addition, each figure should be described in the results section; as currently outlined, several of the figures are included haphazardly without intent to argue a point and strengthen the original purpose of the study. Consider revising the discussion section to "connect-the-dots" for the reader with regard to the significant findings of each figure and how they provide a collective answer to the original question(s). Another suggestion would be reducing the number of figures by summarizing the data in a table format - still need to describe in results and elaborate in discussion sections. 

Overall: relevance and unique findings are clear. However, as currently written, the manuscript is unclear and disjointed. The authors include data that is not adequately described or not described at all in the context of the entire scope of the study. Several other irregularities remain that are described above. 

Comments on the Quality of English Language

Moderate editing of the English language required. 

Author Response

Review 3

Introduction: As currently written, it remains unclear how prevalence and hospital protocol (i.e., precautionary measures, spread of infection etc.) are linked. 

See p7; lines 178 - 188

Consider expanding the introduction to describe the current clinical strategy for identifying C. auris and subsequent protocols for mitigating the spread of infection. What was/is the strategy for c. haemulonii, which according to the authors, is the organism misdiagnosed when evaluating for C. auris. How do these clinical strategies differ between different Arab nations (i.e., Saudi Arabia and The United Arab Emirates). 

Addressed in the Introduction

See also, Pp 5-6; Lines: 125 - 148

Methods: clearly describe what software and/or protocols were used to retrospectively analyze the data. As currently written, it remains unclear if all the samples originated from Saudi Arabia or elsewhere. Remains unclear why only the first isolate per patient was included. Also, why were there yeast cultures performed; is the susceptibility data included? It is difficult to identify the corresponding data to each assay; indicating the figure or table after each method or result would be helpful. 

Only first Positive sample are considered to avoid duplicity

All samples were taken in our hospital

Issues of figures/tables have been addressed. Irrelevant ones removed, and better described in the text.

Results. consider including a more thorough description for each table and figure. As currently written, it is unclear why clinical versus surveillance samples is relevant to the original question and intent of the study. Need to describe all of the data in each figure. Seems authors selected only a subset of reported data to describe at length in the results section. 

Fully addressed. See e.g., clinical versus surveillance samples

Pp 12 – 13; lines 321 - 335

Discussion: Perhaps the weakest portion of the manuscript. Several of the figures are not mentioned at all in the discussion and not referenced in text. Either expand the discussion to include all of the figures/data or remove several figures that are not mentioned or discussed. In addition, each figure should be described in the results section; as currently outlined, several of the figures are included haphazardly without intent to argue a point and strengthen the original purpose of the study. Consider revising the discussion section to "connect-the-dots" for the reader with regard to the significant findings of each figure and how they provide a collective answer to the original question(s). Another suggestion would be reducing the number of figures by summarizing the data in a table format - still need to describe in results and elaborate in discussion sections. 

Addressed: Further expanded to discuss results, and comparing it with other studies’ findings

See pp 14 17

Round 2

Reviewer 1 Report

Comments and Suggestions for Authors

I could not review the manuscript considering that there are errors in the order of the references and when a contrast is made between these and the cited text, there is no coincidence. In case you require my review, you should first send a proper version.

Added to the above there is the following text (Line 243 and 255): Error! Reference source not found.).

There is text in methods that does not correspond to the study: "The multiparametric approach may have been successfully used in some research, e.g., to identify high-risk prostate cancer (with a Gleason score of at least 7) with better sensitivity and specificity than that provided by PSA screening alone.[55] ".

It is not appropriate for the authors to use the 95% CI when it is a descriptive study. This would be correct only if they used random sampling.

Comments on the Quality of English Language

Moderate editing of English language required

Reviewer 3 Report

Comments and Suggestions for Authors

Sufficiently improved

review minor grammatical errors

Comments on the Quality of English Language

Review minor grammatical errors